# Biosynthesis, Chemistry, and Pharmacology of Polyphenols from Chinese *Salvia* Species: A Review

**DOI:** 10.3390/molecules24010155

**Published:** 2019-01-02

**Authors:** Jie Wang, Jianping Xu, Xue Gong, Min Yang, Chunhong Zhang, Minhui Li

**Affiliations:** 1Inner Mongolia Research Center of Characteristic Medicinal Plants Cultivation and Protection Engineering Technology, Baotou Medical College, Baotou 014060, Inner Mongolia, China; yiuje_sgnaw@outlook.com (J.W.); xu_jianping1992@163.com (J.X.); gongxue_2017@yeah.net (X.G.); yangmin_0406@aliyun.com (M.Y.); 2Inner Mongolia Institute of Traditional Chinese Medicine, Hohhot 010020, Inner Mongolia, China

**Keywords:** biosynthesis, chemistry, polyphenols, phenolic acids, pharmacology, *Salvia*

## Abstract

*Salvia* species find widespread application in food and pharmaceutical products owing to their large polyphenol content. The main polyphenols in Chinese *Salvia* species are phenolic acids and flavonoids, which exhibit anti-oxygenation, anti-ischemia-reperfusion injury, anti-thrombosis, anti-tumour, and other therapeutic effects. However, there are few peer-reviewed studies on polyphenols in Chinese *Salvia* species, especially flavonoids. This review is a systematic, comprehensive collation of available information on the biosynthesis, chemistry, and pharmacology of Chinese *Salvia* species. We believe that our study makes a significant contribution to the literature because this review provides a detailed literary resource on the currently available information on various polyphenolic components of Chinese *Salvia* species, including their bioactivities and structures. In addition, the study provides information that would encourage further investigation of this plant material as a natural resource with potential for a broad range of applications in various industries, such as the food and pharmaceutical industries.

## 1. Introduction

The genus *Salvia* is a dominant genus of the Lamiaceae family, which is extensively distributed in China (84 species). The primary phytochemical constituents of Chinese *Salvia* species include sesquiterpenes (e.g., plebeiolide C and trijugins A), diterpenoids (e.g., tanshinone IIA and dihydrotanshinone I), triterpenoids (e.g., 1β,11α-dihydroxyolean-18-en-3-one and 1β,11α,20-trihydroxy-lupan-3-one) [1], phenolic acids (e.g., 3,4-dihydroxycinnamicacid (caffeic acid) and 3-[3,4-dihydroxyphenyl]lactic acid (danshensu)) [2], flavonoids (e.g., luteolin and gardenin-B) [3], alkaloids (e.g., isosalvamines C and salvianan) [4], and saccharides [5]. The main bioactive ingredients in *Salvia* species are polyphenolic compounds (phenolic acids and flavonoids). Phenolic acids have diverse structures (monomers, dimers, trimers, tetramers, and multimers) and are present in high concentrations, which has important taxonomic significance for Chinese *Salvia* species [2]. For example, the monomer caffeic acid is the basic phenolic acid compound. The caffeic acid dimer, rosmarinic acid, is considered the chemotaxonomic marker at the subfamily level in Lamiaceae [6]. In addition, phenolic acids mediate the major pharmacological activities of Chinese *Salvia* species, such as anti-oxygenation [7], anti-ischemia-reperfusion injury [8], and anti-thrombosis [9] effects. Owing to their high nutritional value, Chinese *Salvia* species play an important role in the food [10,11], pharmaceutical [12,13], and cosmetic industries [14].

Polyphenols are a class of important secondary metabolites with multiple phenolic hydroxyl groups, and include flavonoids, phenolic acids, stilbenes, and tannins (hydrolysable and condensed) [15], which are mainly synthesized by the phenylpropanoid metabolic pathway [16]. They possess various pharmacological activities, such as anti-cardiovascular [17], anti-oxidation [18], anti-inflammatory [19], and anti-tumour [20] effects, and are widely distributed among Chinese *Salvia* species [21]. There has been increasing attention on research and development of Chinese *Salvia* species because of the nutritional potential of polyphenols.

This review focuses on the chemical and pharmacological properties of polyphenols in Chinese *Salvia* species. Moreover, it highlights the biosynthetic pathways of polyphenols.

## 2. Biosynthetic Pathway of Phenolic Acids

The biosynthetic pathways of polyphenols include the shikimic acid and phenylpropanoid metabolism pathways [22]. The polyphenols in *Salvia* species are mainly produced by the phenylpropanoid metabolic pathway [16,23,24], and most derivatives have similar basic structures [25]. Phenylalanine and tyrosine are precursor compounds of the phenylpropanoid metabolic pathway, and their biosynthetic pathways constitute two parallel branches of this pathway, which involve five rate-limiting enzymes [26,27]. The enzymes are phenylalanine ammonia-lyase (PAL; a key regulatory enzyme in plant metabolism), cinnamic acid-4-hydroxylase (C4H) and 4-coumarate: coenzyme A (CoA) ligase (a key regulatory enzyme in the phenylalanine branch), tyrosine aminotransferase (the first key enzyme and rate-limiting enzyme in the tyrosine metabolism pathway), and rosmarinic acid synthase (a key enzyme in catalytic synthesis) [28,29].

The phenolic acids of *Salvia* species are caffeic acid derivatives, which are mostly formed by esterification of caffeic acid with danshensu [2,30]. Caffeic acid belongs to the class of phenylpropionic acids [31], and is the basic structural unit of phenolic acids [32]. The precursor compound of caffeic acid is phenylalanine, which produces caffeic acid through the action of PAL and C4H enzymes. It plays a central role in the phenylpropanoid metabolic pathway, and is a precursor compound of rosmarinic acid [28]. Studies have speculated that, in the main synthetic route of rosmarinic acid, caffeic acid is first catalysed to caffeoyl CoA. Subsequently, caffeoyl CoA and 4-hydroxyphenyllactic acid are catalysed by hydroxycinnamoyl-CoA: hydroxyphenyllactate hydroxycinnamoyl transferase (rosmarinic acid synthase (RAS)) to generate caffeoyl-4′-hydroxyphenyllactic acid (caffeoyl-4′-HPLA). Finally, it is catalysed by CYP98A14 to rosmarinic acid [33]. Salvianolic acid E is formed by rosmarinic acid under the action of enzymes and other reactions, and then transformed into salvianolic acid B and other compounds. This observation suggests that rosmarinic acid is the core constituent unit of a series of complex phenolic acids, such as salvianolic acids [34,35,36]. The synthesis of rosmarinic acid has great significance for the formation of complex phenolic compounds [25].

Moreover, the phenylpropanoid metabolic pathway is an important upstream pathway for producing flavonoids (e.g., anthocyanins, flavonoids, and isoflavonoids) [37,38,39,40]. The biosynthetic pathway of phenolic acids in *Salvia* species is shown in Figure 1.

## 3. Chemical Constituents and Structure of Polyphenols

The polyphenolics in Chinese *Salvia* species are phenolic acids, flavonoids, and anthocyanins [41]. Phenolic acids are one of the main active components [42].

### 3.1. Phenolic Acids

Caffeic acid and danshensu are the structural units of phenolic acids in *Salvia* species [2,43]. Phenolic acids can be classified as caffeic acid monomers, dimers, trimers, tetramers, and multimers based on their polymerization degree [2,44,45,46]. Phenolic acids and their representative structures are shown in Table 1 and Figure 2, respectively.

Caffeic acid monomers mainly consist of caffeic acid, danshensu, protocatechuic acid, and protocatechuic aldehyde. Caffeic acid is the basic compound of caffeic acid monomers [32]. Danshensu is also a basic component of caffeic acid derivatives in plant metabolites and the hydrolysate of caffeic acid [2,3,43,46,47]. Rosmarinic acid and salvianolic acids D, F, and G are caffeic acid dimers [43]. Rosmarinic acid is the simplest caffeic acid dimer, which was first isolated from *Rosmarinus officinalis* L. in 1958 [48]. It has a high taxonomic significance in *Salvia* species [6]. Salvianic acid C is synthesized by the condensation of two molecules of caffeic acid [3,36]. Caffeic acid trimers comprise the following members: salvianolic acid A, lithospermic acid, and yunnaneic acid C [2,3]. Structurally, salvianolic acid A has a similar structure to that of salvianolic acid F. It has been speculated that salvianolic acid A is the product of salvianolic acid F and danshensu synthesis [43,47,49]. Lithospermic acid is a typical trimer that is broadly distributed in Chinese *Salvia* species [6]. Caffeic acid tetramers can be regarded as derivatives of rosmarinic acid dimers [43]. They mainly contain salvianolic acid B (typical tetramer), salvianolic acid E, and yunnaneic acid G [25,36,47]. Salvianolic acid B has potential taxonomic value in *Salvia* species and represents the main active constituent [6]. Yunnaneic acid A and B are the caffeic acid multimer extracted from the roots of *Salvia yunnanensis* C. H. Wright. They are hexamers of caffeic acid and regarded as products of the combination of caffeic acid and rosmarinic acid. Yunnaneic acid B contains two units of yunnaneic acid C, while yunnaneic acid A is a combination of yunnan acid C and D [50,51].

Presently, three compounds of phenolic acid salts, namely sodium danshensu, magnesium lithospermate B, and ammonium-potassium lithospermate B, have been detected in *Salvia* species [43]. Magnesium lithospermate B and ammonium-potassium lithospermate B are magnesium and ammonium-potassium compounds of the tetramer salvianolic acid B. Magnesium lithospermate B has antioxidative [52], anti-liver injury [53], and anti-myocardial ischemia-reperfusion injury effects [54].

### 3.2. Other Compounds

In addition to phenolic acids, other phenolic compounds, such as flavonoids and anthocyanins, are also present in *Salvia* species [100,101,102]. Based on the biosynthesis pathway of phenolic acids, flavonoids are known to be produced through the phenylpropanoid pathway [37,38]. Currently, flavonoids are isolated from natural plants, such as *Salvia plebeia* R. Br. and *Salvia miltiorrhiza* Bunge [103]. Furthermore, research studies have reported the total flavonoid content in *Salvia cavaleriei* Levl. to be 14.69% [104].

Moreover, *Salvia* species contain anthocyanin and tannin compounds. The compounds salvianin and monardalins are obtained from *Salvia coccinea* L. and *Salvia splendens* Ker-Gawl., which are good natural spices [105,106]. The flavonoids in *Salvia* species are shown in Table 2 and Figure 3.

## 4. Pharmacological Activities of Phenolic Acids

Phenolic acids are the main active polyphenols in Chinese *Salvia* species with an active function in promoting health. Phenolic acids are rich in phenolic hydroxyl groups. Accordingly, representative compounds have been reported to possess a wide variety of activities, including anti-oxygenation [7], anti-ischemia-reperfusion injury [8], and anti-thrombosis [9] effects.

### 4.1. Anti-Oxygenation Activity

Oxidative damage is mainly caused by substances, such as free radicals, present inside and outside the cell. The inability to remove a large amount of free radicals in a timely manner causes dynamic imbalances in the redox system of the body. The antioxidant effects of phenolic acids involve three processes: free radical scavenging, inhibiting free radical generation, and anti-lipid peroxidation [8,122,123].

Phenolic acids possess radical scavenging activity. A study showed that, at 0–0.7 mmol·L^−1^, rosmarinic acid, caffeic acid, and danshensu were as effective as quercetin (positive control), which scavenged 2,2-diphenyl-1-picrylhydrazyl (DPPH) radicals in a concentration-dependent manner. In contrast, ferulic acid was less effective [124]. Moreover, other studies found that salvianolic acid B and danshensu exhibited higher scavenging activities against HO·, O_2_^−^, DPPH, and 2,2′-azino-*bis*(3-ethylbenzothiazoline-6-sulphonic acid (ABTS) than the other constituents. Among them, the half-maximal inhibitory concentration (IC_50_) of danshensu and salvianolic acid B were 94 and 102 μg·mL^−1^, respectively, and there was no obvious difference in the scavenging ability of hydroxyl radicals [7]. Thus, phenolic acids are considered to be hydroxyl radical scavengers.

The antioxidant activities of rosmarinic acid and salvianolic acid B in plant extracts were determined using DPPH radical scavenging, superoxide radical quenching, a β-carotene-linoleic acid system, and reductive potential assays. The data indicated that salvianolic acid B and rosmarinic acid, which were abundantly found in *Salvia*
*miltiorrhiza* leaves, had strong antioxidant activity. Among them, salvianolic acid B was positively correlated with DPPH scavenging activity (correlation coefficient (*r*) = 0.693, *P* < 0.05), reducing ability (*r* = 0.748, *P* < 0.01), and inhibition of linoleic acid oxidation (*r* = 0.804, *P* < 0.01). It could be regarded as a new possible resource of natural phenolic antioxidants [125].

Studies have shown that the order of inhibition of spontaneous lipid peroxidation in liver tissue of mice by the following constituents is rosmarinic acid (minimum inhibitory concentration (MIC): 12.5 μg·mL^−1^, 54.5%), protocatechuic aldehyde (50.1%), chlorogenic acid (46.9%), caffeic acid (MIC: 50.0 μg·mL^−1^, 38.4%), ferulic acid (MIC: 50.0 μg·mL^−1^, 36.1%), danshensu (MIC: 50.0 μg·mL^−1^, 36.3%), and 3-hydroxycinnamic acid (32.2%). However, the protective effects of various constituents on hydrogen peroxide (H_2_O_2_)-induced liver lipid peroxidation in mice showed the following values: rosmarinic acid (MIC: 49.9 μg·mL^−1^, 99.0%), caffeic acid (98.9%), protocatechuic aldehyde (98.3%), chlorogenic acid (90.0%), ferulic acid (83.2%), danshensu (54.3%), and 3-hydroxycinnamic acid (42.1%). At the same time, the total radical reducing activities of the seven phenolic acids were determined to be consistent with the above results [126].

### 4.2. Anti-Myocardial/Cerebral Ischemia-Reperfusion Injury Activity

The phenolic acids in *Salvia* species are water-soluble components that are active against cardiovascular diseases. The pathophysiological mechanism of myocardial/cerebral ischemia-reperfusion injury is complex, and phenolic acids act against it through mechanisms such as regulation of active oxygen metabolism, inhibition of inflammatory reaction, apoptosis, and calcium overload [2,8].

Salvianolic acid B was found to effectively reduce myocardial ischemia-reperfusion injury. The experiment established an ischemia-reperfusion model by ligating the left circumflex artery in Sprague–Dawley (SD) rats. The effects of different doses of salvianolic acid B (20, 40, and 60 mg·d^−1^·kg^−1^) against myocardial ischemia-reperfusion injury were determined by measuring the concentration and apoptotic index of the plasma level of myocardial enzymes (cardiac troponins (CTn) I and creatine kinase-MB (CKMB)), malondialdehyde (MDA), endothelin (ET), nitric oxide (NO), and superoxide dismutase (SOD), and histological changes of the heart. The effects were superior in the salvianolic acid B high-dose group (60 mg·d^−1^·kg^−1^). The results showed that salvianolic acid B significantly increased the plasma CTn I, CKMB, MDA, and ET contents; decreased NO and T-SOD contents; reduced the infarct size; and improved myocardial ultrastructure changes. The study showed that salvianolic acid B protected against conditions such as myocardial ischemia-reperfusion injury by regulating active oxygen metabolism, reducing oxidative stress, and myocardial apoptosis [127].

Salvianolic acid A protected rats against cerebral ischemia-reperfusion injury by inhibiting the expression of matrix metalloproteinase (MMP)-9 and inflammatory response. Cerebral ischemia-reperfusion injury significantly upregulates the expression of MMP-9, leading to severe damage of the blood–brain barrier. Furthermore, it activates inflammatory response, produces oxygen free radicals, or releases lysosomal enzymes to damage tissue. Studies have shown that the level of MMP-9 in salvianolic acid A-treated groups (5, 10, and 20 mg·kg^−1^) was downregulated, tissue inhibitor of metalloproteinase-1 (TIMP-1) was upregulated, and damage to the blood–brain barrier was reduced. Salvianolic acid A inhibits activation of cerebral nuclear factor (NF)-κB p65 and reduces inflammatory response. Salvianolic acid A could be used as an effective drug to alleviate cerebral ischemia-reperfusion injury [128]. In addition, it can prevent myocardial ischemia-reperfusion injury under a high glucose condition by regulating the NADPH oxidase 2 (Nox2)/reactive oxygen species (ROS)/phosphorylated-c-Jun N-terminal kinase 2 (p-JNK2)/NF-κB pathway to reduce transient receptor potential cation channel, subfamily C, member 6 (TRPC6)/Ca^2+^ influx [129].

### 4.3. Anti-Thrombosis Activity

Thrombosis can occur anywhere in the blood circulation, causing acute myocardial infarction, cerebral infarction, and pulmonary thrombosis [130]. Presently, the strategies for preventing thrombosis mainly include improving blood rheology, anti-platelet aggregation, and protecting vascular endothelial cells [131].

Studies have shown that salvianolic acids in *S.*
*miltiorrhiza* have anti-thrombotic effects and they act primarily by improving blood rheology, anti-platelet aggregation, and targeting P2Y_1_ or P2Y_12_ receptors, which are new target receptors for anti-platelet aggregation. It can be seen from the experimental results that salvianolic acid B (100 µmol·L^−1^) only antagonizes the action of platelet P2Y_12_ receptors, while salvianolic acid A and C (both 100 µmol·L^−1^) are P2Y_1_ and P2Y_12_ receptor inhibitors [9].

Caffeic acid can inhibit platelet-mediated thrombosis. High activation of platelets is one of the major causes of thrombosis. The adhesion and aggregation of platelets are enhanced in the activated state. The results demonstrated that caffeic acids have beneficial effects on aberrant platelet activation-related diseases. Caffeic acid (25–100 μmol·L^−1^) repressed ADP-induced platelet aggregation, P-selectin expression, ATP release, and Ca^2+^ mobilization. Furthermore, it attenuates the activation of p38, ERK, integrin αIIbβ_3_, and JNK. It could also increase the expression level of cAMP [132].

Danshensu (15–60 mg·kg^−1^) had better anti-thrombotic and antiplatelet therapeutic efficacy than other constituents. It mainly contributed to intensely and selectively suppressing the expression of cyclooxygenase (COX)-2 and balancing the ratio of thromboxane A2 (TXA2)/prostacyclin (PGI2) [133].

### 4.4. Anti-Liver Injury Activity

Liver damage is a common pathological manifestation of various liver diseases, and further progression leads to different degrees of hepatic cell necrosis, fatty liver, liver cirrhosis, and other liver diseases. Phenolic acids are potential anti-liver damage compounds. Liver protection can be achieved through the following mechanisms: elimination of free radicals, inhibition of lipid peroxidation, and inhibition of inflammatory cytokine expression [134,135].

Salvianolic acid B (15 or 30 mg·d^−1^·kg^−1^) protects liver cells by preserving lysosomal membrane integrity through scavenging ROS and enhancing lysosome-associated membrane protein 1 (LAMP1) expression. LAMP1 acts as a barrier to soluble hydrolase, preventing cell damage and death caused by the release of luminal contents into the cytoplasm. H_2_O_2_ attack significantly decreases LAMP1 expression, thereby disrupting the integrity of hepatocyte lysosomal membrane, leading to cell damage and death. Liver cells are protected by the anti-inflammatory and anti-oxidant properties of salvianolic acid B [136].

Salvianolic acid A protects against acute hepatic injury induced by concanavalin A (ConA) in mice. The results of the liver function indicators serum aspartate aminotransferase (AST) and alanine aminotransferase (ALT) showed that salvianolic acid A (15 or 25 mg·kg^−1^) significantly reduced ConA-induced ALT and AST activity. In addition, it reduced hepatotoxic cytokine levels, such as tumour necrosis factor (TNF)-α and interferon (IFN)-γ; ameliorated the increased NF-κB level and cleaved caspase-3; and reversed B-cell lymphoma-extra-large (Bcl-xL) expression. Notably, pretreatment with salvianolic acid A obviously upregulated the expression of SIRT1, which could alleviate acute hypoxic injury and metabolic liver disease. The study showed that the increase in SIRT1 was closely related to the p66 isoform (p66shc) of the growth factor adapter Shc. Other studies have shown that salvianolic acid A-alleviated ConA-induced hepatitis may be inhibited by the SIRT1-mediated p66shc pathway [137].

In addition, studies have shown that the water extract of *S.*
*miltiorrhiza* (0.06–1 mg·mL^−1^) has the same hepatoprotective effect as salvianolic acid A does, and the mechanism may be related to the reduction of extracellular histone and related cytokines. A water extract of *S.*
*miltiorrhiza* contained danshensu (8.2–130.5 μmol·L^−1^) and salvianolic acid B (3.3–53.5 μmol·L^−1^) as the main compounds that protected against liver injury. However, rosmarinic acid, protocatechuic aldehyde, and salvianolic acid A had no protective effects [138].

### 4.5. Anti-Tumour Activity

Phenolic acids have multitarget anti-tumour effects; however, their mechanisms are more complicated. They prevent the invasion of tumour cells by inducing apoptosis of tumour cells, regulating immune function, reversing multidrug resistance of cancer cells, targeting tumour microtubules to inhibit their proliferation and division, and inhibiting metastasis of cancer cells [139,140].

Salvianolic acid B inhibits the growth of human glioma U80 cells, which may be involved in p38-activation-mediated ROS production. These experimental results indicate that salvianolic acid B (1–100 μmol·L^−1^) significantly reduced the cell viability of U80 cells in a dose- and time-dependent manner. At the same time, it enhanced the production of intracellular ROS and induced apoptosis of U57 cells. The anti-tumour activity of salvianolic acid B in vivo was observed in a nude mouse xenograft model. Therefore, salvianolic acid B seems to be safe and effective, and this natural component could be developed into a potential therapeutic agent for glioma [141].

### 4.6. Others

Phenolic acids in *Salvia* species exert anti-hypertensive effects [142,143], improve memory and cognitive impairment [144,145], possess hypoglycaemic [146,147] and antiviral activities [148,149], and can be used to prevent and treat cataract [150,151]. These pharmacological activities of phenolic acids in Chinese *Salvia* species are listed in Table 3.

## 5. Conclusions

In recent years, the antioxidant properties of polyphenolic compounds have encouraged their widespread used as supplements in food or pharmaceutical products. *Salvia* species are a rich source of polyphenols, whose application has been increasing widely used in many countries. The main polyphenolic compounds are phenolic acids (rosmarinic acid, salvianolic acid, and their derivatives), which are based on caffeic acid with compounds formed from two to four or more caffeic acid units. Most of the biological activities, such as anti-oxidant, anti-ischemia-reperfusion, and anti-thrombosis effects are attributed to phenolic acids. Nevertheless, in the study of the pharmacological mechanism of phenolic acids, detailed experiments have been conducted only on common compounds, such as rosmarinic acid, salvianolic acid A, and salvianolic acid B, whereas other compounds are less studied.

The flavonoids found in Chinese *Salvia* species are apigenin, acacetin, and luteolin. However, few studies have been conducted on flavonoids, including their phytochemistry and pharmacological effects. This is a shortcoming in the study of chemical constituents and pharmacological effects of Chinese *Salvia* species. The flavonoids in Chinese *Salvia* species should be analysed using modern analytical techniques, such as liquid chromatography-tandem mass spectrometry to supplement current knowledge on the chemical composition of phenolic acids.

Investigating the biosynthesis of polyphenols is a challenging task in Chinese *Salvia* species. The medicinal and economic values of phenolic acids make it particularly important to increase the content of phenolic acids in Chinese *Salvia* species. Presently, only the synthesis of upstream rosmarinic acid has been partially elucidated in the biogenic pathways of phenolic acids. The source pathways of other phenolic acids, such as salvianolic acid B downstream of rosmarinic acid, have not yet been elucidated. Therefore, elucidation of the conversion pathway of phenolic acid, particularly identifying key enzymes involved in transformation, and in-depth study of the biotransformation mechanism would promote the production and development of new phenolic acids.

Owing to the extensive use of Chinese *Salvia* species in food and medicine, it is necessary to explore polyphenols more comprehensively. In this review, we systematically summarized the biosynthesis, phytochemistry, and pharmacology of Chinese *Salvia* species, which provide a basis for its further development and utilization as a resource.

## Figures and Tables

**Figure 1 molecules-24-00155-f001:**
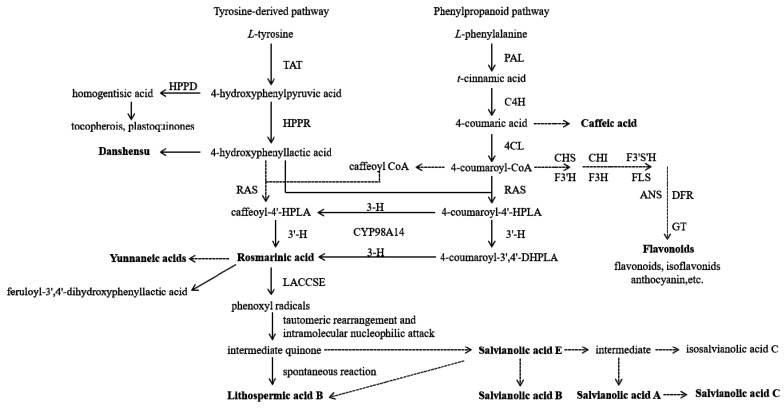
The biosynthetic pathway of phenolic acids in *Salvia* species. TAT: tyrosine aminotransferase; HPPR: 4-hydroxyphenylpyruvate reductase; HPPD: 4-hydroxyphenylpyruvated dioxygenase; PAL: phenylalanine ammonia-lyase; C4H: cinnamic acid 4-hydroxylase; 4CL: 4-coumarate-CoA ligase; RAS: hydroxycinnamoyl-CoA: hydroxyphenyllactate hydroxycinnamoyl transferase; caffeoyl-4′-HPLA: caffeoyl-4′-hydroxyphenyllactic acid; 4-coumaroyl-4′-HPLA: 4-coumaroyl-4′-hydroxyphenyllactic acid; 4-coumaroyl-3′,4′-DHPLA: 4-coumaroyl-3′,4′-dihydroxyphenyllactic acid; 3-H: hydroxycinnamoyl-hydroxyphenyllactate 3-hydroxylase, 3′-H: hydroxycinnamoyl-hydroxyphenyllactate 3′-hydroxylase. Solid line: the verified biosynthesis process; dotted line: proposed biosynthesis processes.

**Figure 2 molecules-24-00155-f002:**
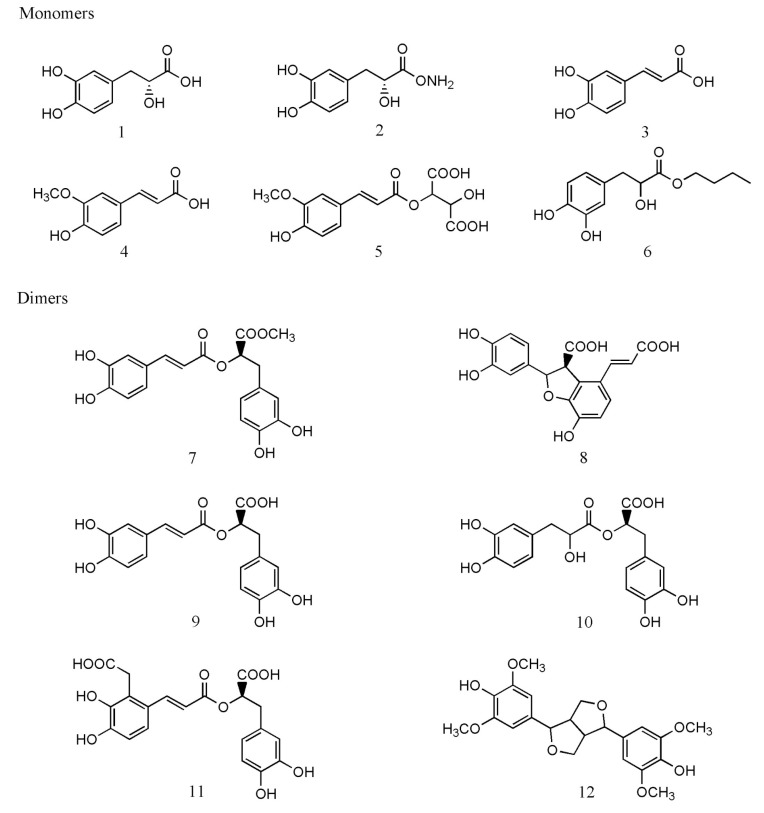
The representative structures of phenolic acids in *Salvia* species.

**Figure 3 molecules-24-00155-f003:**
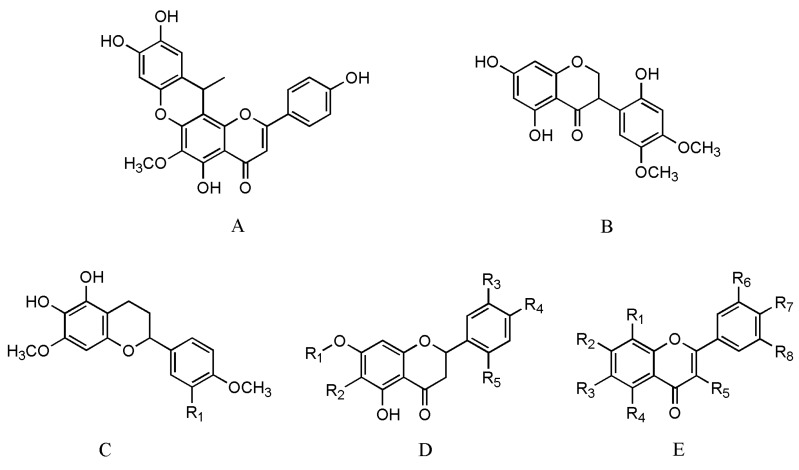
The structures of flavonoids in *Salvia* species.

**Table 1 molecules-24-00155-t001:** The phenolic acids in *Salvia* species.

NO.	Structures	Compound	Species	References
1	Monomers	(2,3,4-trihydroxy-3-methyl)butyl-6-feruloylglucoside	*S. officinalis*	[55]
2		1-caffeoyl-6-apiosyl-glucoside	*S. officinalis*	[56]
3		(3-methoxy-4-glucosyloxyphenyl)-3-hydroxymethyl-5-(3-hydroxypropyl)-7-methoxy-2,3-dihydrobenzofuran	*S. officinalis*	[57]
4		3-(3,4-dihydroxyphenyl)lactic acid(danshensu) (**1**)	*S. miltiorrhiza*	[58]
			*S. chinensis*	[59]
			*S. prionitis*	[60]
			*S. sonchifolia*	[61]
5		3-(3,4-dihydroxyphenyl)lactamide (**2**)	*S. miltiorrhiza*	[62]
6		3,4-dihydroxycinnamic acid(caffeic acid) (**3**)	*S. miltiorrhiza*	[63]
			*S. bowleyana*	[63]
			*S. chinensis*	[59]
			*S. officinalis*	[55]
			*S.* *plebeia*	[64]
			*S. sonchifolia*	[61]
7		3-methoxy-4-hydroxybenzoic acid (vanillic acid)	*S. officinalis*	[65]
8		4-hydroxyacetophenone 4-(2-(5-syringoyl)apiosyl)glucoside	*S. officinalis*	[55]
9		4-hydroxybenzoic acid	*S. officinalis*	[66]
10		4-hydroxyphenyllactic	*S. plebeia*	[65,67]
11		6-caffeoyl-1-fructosyl-*a*-glucoside	*S. officinalis*	[56]
12		6-feruloyl-*a*-glucose	*S. officinalis*	[57]
13		6-feruloyl-*b*-glucose	*S. officinalis*	[57]
14		ailanthoidol	*S.miltiorrhiza*	[68]
15		coniferyl aldehyde	*S. plebeia*	[69]
16		ferulic acid (**4**)	*S. officinalis*	[70]
17		isoferulic acid	*S. miltiorrhiza*	[70]
18		methyl 3,4-dihydroxyphenyllactate	*S. plebeia*	[71]
19		*m*-hydroxybenzaldehyde	*S. przewalskii*	[72]
20		*mono*-feruloyltartaric acid (**5**)	*S. chinensis*	[64]
21		*n*-butyl 3,4-dihydroxyphenyllactate (**6**)	*S. plebeia*	[71]
22		*p*-hydroxycinnamic acid	*S. miltiorrhiza*	[73]
23		prionitiside A	*S. prionitis*	[60]
24		prionitiside B	*S. prionitis*	[74]
25		protocatechuic acid	*S. miltiorrhiza*	[75]
			*S. sonchifolia*	[55]
26		protocatechuic aldehyde	*S. miltiorrhiza*	[75]
27		salvinal	*S.miltiorrhiza*	[76]
28		salviaplebeiaside	*S. plebeia*	[77]
29	Dimers	1-hydroxypinoresinol 1-glucoside	*S. officinalis*	[57]
30		2-(3-methoxy-4-hydroxyphenyl)-5-(3-hydroxypropyl)-7-methoxybenzofuran-3-carbaldehyde	*S. miltiorrhiza*	[68,78]
31		isolariciresinol 3α-glucoside	*S. officinalis*	[79]
32		isolariciresinol di(12-methylmyristate)	*S. plebeia*	[80]
33		isosalvianolic acid C	*S. cavaleriei*	[59]
34		methyl rosmarinate (**7**)	*S.miltiorrhiza*	[56]
			*S. bowleyana*	[63]
			*S. prionitis*	[60]
35		prolithospermic acid (przewalskinic acid A) (**8**)	*S.miltiorrhiza*	[2]
36		rosmarinic acid (**9**)	*S. miltiorrhiza*	[81]
37		salvianic acid C (**10**)	*S. miltiorrhiza*	[2]
38			*S. cavaleriei*	[60]
			*S. bowleyana*	[63]
			*S. prionitis*	[60]
		salvianolic acid D (**11**)	*S. miltiorrhiza*	[70]
39			*S. chinensis*	[74]
		salvianolic acid F	*S.miltiorrhiza*	[2]
40		salvianolic acid G	*S.miltiorrhiza*	[2]
41		salvianolic acid N	*S. yunnanensis*	[82]
42		salviaflaside	*S. flava*	[82]
43		salviaflaside methyl ester	*S. flava*	[83]
44		Syringaresinol (**12**)	*S. plebeia*	[84]
45	Trimers	9″-methyl lithospermate	*S. miltiorrhiza*	[81]
46		*cis*-lithospermic acid	*S. yunnanensis*	[85]
47		dimethyl lithospermate	*S. miltiorrhiza*	[81]
48		ethyl lithospermate	*S.miltiorrhiza*	[70]
49		ethyl salvianolate A	*S. yunnanensis*	[60]
50		feruloylisolariciresinol 12-methylmyristate	*S. plebeia*	[86]
51		lithospermic acid (**13**)	*S. miltiorrhiza*	[81]
52		lithospermic acid dimethyl ester	*S. miltiorrhiza*	[81]
53		lithospermic acid monomethyl ester	*S. miltiorrhiza*	[81]
54		methyl salvianolate A	*S. yunnanensis*	[85]
55		methyl salvianolate I	*S. officinalis*	[79]
56		methyl salvianolic acid C	*S.miltiorrhiza*	[87]
57		sagecoumarin	*S. officinalis*	[79]
58		salvianolic acid A (**14**)	*S. miltiorrhiza*	[88]
59			*S. cavaleriei*	[60]
			*S. flava*	[89]
			*S. yunnanensis*	[85]
		salvianolic acid C	*S. miltiorrhiza*	[90]
60		salvianolic acid H	*S. cavaleriei*	[91]
61		salvianolic acid I	*S. officinalis*	[79]
62			*S. cavaleriei*	[89]
		salvianolic acid J	*S. flava*	[92]
63		salvianolic acid K	*S. deserta*	[79]
64		salvianolic acid T	*S. miltiorrhiza*	[82]
65		salvianolic acid U	*S. miltiorrhiza*	[82]
66		yunnaneic acid C	*S. yunnanensis*	[51,93]
67		yunnaneic acid D	*S. yunnanensis*	[51,93]
68		yunnaneic acid E	*S. yunnanensis*	[94]
69		yunnaneic acid F	*S. yunnanensis*	[94]
70	Tetramers	9′-monomethyl lithospermate B	*S. przewalskii*	[95]
71		9‴-monomethyl lithospermate B	*S. przewalskii*	[95]
72			*S. miltiorrhiza*	[96]
		dimethyl lithospermate B	*S. przewalskii*	[95]
73			*S. miltiorrhiza*	[96]
		ethyl lithospermate B	*S. miltiorrhiza*	[70]
74		Rabdosiin (**15**)	*S. yunnanensis*	[97]
75		sagerinic acid	*S. officinalis*	[79]
76		salvianolic acid B(lithospermic acid B) (**16**)	*S. miltiorrhiza*	[90]
77		salvianolic acid E	*S. miltiorrhiza*	[70]
78		salvianolic acid L	*S. officinalis*	[98]
79		yunnaneic acid G	*S. yunnanensis*	[94]
80		yunnaneic acid H	*S. yunnanensis*	[94]
81	Multimers	yunnaneic acids A (**17**)	*S. yunnanensis*	[51,93]
82		yunnaneic acid B	*S. yunnanensis*	[51,93]
83	Salts	ammonium-potassium lithospermate B	*S. miltiorrhiza*	[99]
84		sodium danshensu	*S. miltiorrhiza*	[73]
85		magnesium lithospermate B (**18**)	*S. miltiorrhiza*	[99]

**Table 2 molecules-24-00155-t002:** The flavonoids in *Salvia* species.

NO.	Compound	Structures	R_1_	R_2_	R_3_	R_4_	R_5_	R_6_	R_7_	R_8_	Species	References
86	neocafhispidulin	A									*S. plebeia*	[69]
87	2′-hydroxy-5′-methoxybiochanin A	B									*S. plebeia*	[69]
88	5,6-dihydroxy-7,4′-dimethoxyflavone	C	H								*S. plebeia*	[107]
89	5,6,3′-trihydroxy-7,4′-dimethoxyflavone		OH								*S. plebeia*	[108]
90	5,7,3′-trihydroxy-4′-methoxyflavanone (hesperetin)	D	H	H	OH	OCH_3_	H				*S. officinalis*	[65]
91	6-methoxynaringenin		H	OCH_3_	H	OH	H				*S. plebeia*	[107]
92	5,3′-dihydroxy-7,4′-dimethoxyflavanone		CH_3_	H	OH	OCH_3_	H				*S. miltiorrhiza*	[109]
93	(2*S*)-5,7,4′-trihydroxy-6-methoxy-flavanone-7-*O-*β-*D*-glucopyran-oside		Glc	OCH_3_	H	OH	H				*S. plebeia*	[108]
94	5,7,3′,4′-tetrahydroxy-6-methoxy-flavanone-7-*O-*β-*D*-glucopyran-oside		Glc	OCH_3_	OH	OH	H				*S. plebeia*	[108]
95	5,6,7,4′-tertrahydroxyflavone	E	H	H	H	OH	H	H	OH	H	*S. plebeia*	[110]
96	luteolin		H	OH	H	OH	H	OH	OH	H	*S. plebeia*	[111]
97	apigenin		H	OH	H	OH	H	H	OH	H	*S. plebeia*	[85]
98	kaempferol		H	OH	H	OH	OH	H	OH	H	*S. roborowskii*	[112]
99	-3′-methyl ether (isorhamnetin)		H	OH	H	OH	OH	H	OH	OCH_3_	*S. farinacea*	[113]
100	quercetin		H	OH	H	OH	OH	OH	OH	H	*S. plebeia*	[114]
101	quercimelin		H	OH	H	OH	Rham	OH	OH	H	*S. roborowskii*	[115]
102	myricitrin		H	OH	H	OH	Rham	OH	OH	OH	*S. roborowskii*	[112]
103	rutin		H	OH	H	OH	Rham-Glc	OH	OH	H	*S. roborowskii*	[115]
104	6-hydroxyapigenin (scutellarein)		H	OH	OH	OH	H	H	OH	H	*S. officinalis*	[65]
105	6-hydroxyluteolin 5-*O*-glucoside		H	OH	OH	OGlc	H	H	OH	OH	*S.* *tomentosa*	[116]
106	hispidulin		H	OH	OCH_3_	OH	H	H	OH	H	*S. plebeia*	[108]
107	pectolinarigenin		H	OH	OCH_3_	OH	H	H	OCH_3_	H	*S. plebeia*	[108]
108	nepetin		H	OH	OCH_3_	OH	H	OH	OH	H	*S. officinalis*	[117]
109	jaceosidin		H	OH	OCH_3_	OH	H	OCH_3_	OH	H	*S. plebeia*	[108]
110	eupatilin		H	OH	OCH_3_	OH	H	OCH_3_	OCH_3_	H	*S. plebeia*	[114]
111	gehkwahin		H	OCH_3_	H	OH	H	H	OH	H	*S. officinalis*	[118]
112	-7,4′-dimethyl ether		H	OCH_3_	H	OH	H	H	OCH_3_	H	*S. officinalis*	[3]
113	kumalakenin		H	OCH_3_	H	OH	OCH_3_	H	OH	H	*S. officinalis*	[118]
114	ayamin		H	OCH_3_	H	OH	OCH_3_	OH	OCH_3_	H	*S. officinalis*	[118]
115	sorbifolin		H	OCH_3_	OH	OH	H	H	OH	H	*S. plebeia*	[108]
116	cirsimaritin		H	OCH_3_	OCH_3_	OH	H	H	OH	H	*S. officinalis*	[118]
117	cirsiliol		H	OCH_3_	OCH_3_	OH	H	H	OH	OH	*S. plebeia*	[107]
118	eupatorin		H	OCH_3_	OCH_3_	OH	H	H	OCH_3_	H	*S. plebeia*	[119]
119	5,6,7,4′-tetramethyl ether		H	OCH_3_	OCH_3_	OCH_3_	H	H	OCH_3_	H	*S. officinalis*	[120]
120	cynaroside		H	OGlc	H	OH	H	OH	OH	H	*S. plebeia*	[114]
121	hispidulin-7-*O*-*D*-glucoside		H	OGlc	OCH_3_	OH	H	H	OH	H	*S. plebeia*	[69]
122	nepitrin		H	OGlc	OCH_3_	OH	H	OH	OH	H	*S. plebeia*	[108]
123	cosmosiin		H	OGlu	H	OH	H	H	OH	H	*S. deserta*	[117]
124	6-hydroxyluteolin-7-glucoside		H	OGlc	OH	OH	H	OH	OH	H	*S. plebeia*	[110]
125	nepetin-7-glucoside		H	OGlu	OCH_3_	OH	H	OH	OH	H	*S. plebeia*	[111]
126	homoplantaginin		H	OGlu	OCH_3_	OH	H	H	OH	H	*S. plebeia*	[108]
127	6″-*O*-acetyl homoplantaginin		H	OGlu-Ac	OCH_3_	OH	H	H	OH	H	*S. plebeia*	[108]
128	luteolin 7-*O*-(4″,6″-di-*O*-α-*L*-rhamnopyranosyl)-β-*D*-glucopyranoside		H	ORham-Glu	H	OH	H	H	OH	OH	*S. splendens*	[121]
129	chrysoeriol-7-*D*-xyloside		H	Oxyloside	H	OH	H	OCH_3_	OH	H	*S. deserta*	[118]
130	salvitin		OH	OCH_3_	H	OH	H	H	OH	H	*S. Plebeia*	[118]

**Table 3 molecules-24-00155-t003:** Other activities of phenolic acids in *Salvia* species

Activity	Ingredient	Model	Treatment	Result	Reference
Anti-hypertension	Caffeic acid, Chlorogenic acid	Cyclosporine-induced hypertensive rats	10 and 15 mg·d^−1^·kg^−1^	Caffeic acid and chlorogenic acid significantly (*p* < 0.05) reduced systolic blood pressure (SBP) and heart rates (HR), activity of angiotensin-1-converting enzyme (ACE), acetylcholinesterase (AChE), butrylcholinesterase (BChE), and arginase in the treated hypertensive rats.	[142]
				Caffeic acid and chlorogenic acid improved nitric oxide (NO) bioavailability, increased catalase activity, and reduced glutathione content while the MDA level was reduced.	
	Salvianolic acid A (SalA)	Spontaneously hypertensive rats (SHR)	2.5, 5, and 10 mg·d^−1^·kg^−1^	The inward remodeling of the retinal vein was inhibited after treatment with SalA.	[143]
				SalA improved the endothelial-dependent vasodilatation of mesenteric vessels for SHR in vivo.	
				Transendothelial electrical resistance (TEER) significantly increased the human umbilical vein endothelial cell line (HUVEC) monolayer treated with SalA.	
				SalA exhibited an obvious protective effect on the HUVEC monolayer.	
Improve memory and cognitive impairment	Rosmarinic acid	Amyloid beta (Aβ) 42-induced echoic memory decline (Rat model of Alzheimer)	50 mg·d^−1^·kg^−1^	It decreased the levels of thiobarbituric acid reactive substances (TBARS) and 4-Hydroxy-2- nonenal (4-HNE) but increased the activity of antioxidant enzymes (superoxide dismutase (SOD), catalase (CAT), and glutathione peroxidase (GSH-Px)) and glutathione levels.	[144]
				Rosmarinic acid (RA) attenuated the increased Aβ staining and astrocyte activation.	
				RA treatment reversed the Aβ42-related alterations in auditory event related potential (AERP) parameters.	
	Total salvianolic acid (TSA)	Alzheimer′s disease modelAPPswe/PS1dE9 mice	30 and 60 mg·d^−1^·kg^−1^, intraperitoneal (i.p.) injection	Treatment with TSA substantially decreased the low-density lipoprotein (LDL)-C level, and 60 mg·kg^−1^ TSA decreased the cholesterol (CHOL) level.	[145]
				The Aβ42 and Aβ40 levels in the hippocampus were decreased.	
Hypoglycemic	Salvianolic acid B (Sal B)	Multiple low-dose streptozotocin (MLDS)-induced diabetes in rat	20 or 40 mg·kg^−1^	They caused a significant decrease of the serum glucose (*p* < 0.05–0.01) and an improvement in the oral glucose tolerance test (OGTT).	[146]
				Serum insulin was significantly higher in Sal B20- and Sal B40-treated diabetics and treatment of diabetics with Sal B40 significantly lowered MDA, raised GSH, and activity of catalase with no significant change of nitrite.	
				The number of pancreatic islets and their area was significantly higher and apoptosis reactivity was significantly lower in the Sal B40-treated diabetic group versus diabetics.	
	Water extract of *S. libanotica*	Animals were fed a high-fat diet	50, 150, and 450 mg·kg^−1^	A decrease in fasting serum glucose and an increase in fasting serum insulin and liver glycogen content.	[147]
				Intake produced a significant improvement in the serum high-density lipoprotein (HDL) and HDL/low-density lipoprotein (LDL) cholesterol ratio, as well as a decrease in abdominal fat.	
Antiviral	Protocatechuic aldehyde	Hepatitis B virus (HBV) replication in the HepG2 2.2.15 cell line	24–48 μg·mL^−1^	Protocatechuic aldehyde appeared to downregulate the secretion of hepatitis B surface antigen (HBsAg) and hepatitis B e antigen (HBeAg) as well as the release of HBV DNA from HepG2 2.2.15 in a dose- and time-dependent manner.	[148]
		Duck hepatitis B virus (DHBV) replication in ducklings	25, 50, or 100 mg·kg^−1^, intraperitoneally, twice daily	Protocatechuic aldehyde also reduced viremia in DHBV-infected ducks.	
	Magnesium lithospermate B	Enterovirus 71 (EV71) viral internal ribosome entry site (IRES)-mediated translation	30 μg·mL^−1^	Magnesium lithospermate B inhibited EV71 infection when they were added to rhabdomyosarcoma (RD) cells during the viral absorption stage.	[149]
				It had a low IC_50_ value of 0.09 mmol·L^−1^ and a high therapeutic index (TI) value of 10.52.	
			100 mg·mL^−1^	Magnesium lithospermate B also reduced EV71 viral particle production and significantly decreased VP1 protein production.	
	Rosmarinic acid	EV71 viral IRES-mediated translation	30 μg·mL^−1^	Rosmarinic acid inhibited EV71 infection when they were added to RD cells during the viral absorption stage.	
				It had an IC_50_ value of 0.50 mmol·L^−1^ and a TI value of 2.97.	
			100 mg·mL^−1^	Rosmarinic acid also reduced EV71 viral particle production and significantly decreased VP1 protein production.	
Prevents and treats cataract	Danshensu	Selenite-induced cataractogenesis in cultured rat lens	500 mmol·L^−1^	Lens morphology: 75% of lenses were transparent, 25% developed only lesser amounts of cortical vacuolization.	[150]
				Danshensu reduces MDA and restores GSH level and total sulfhydryl (SH) content in the lens.	
				Increase of anti-oxidant enzymes (SOD, CAT) activities with danshensu.	
	Protocatechualdehyde	Methylglyoxal-induced mitochondrial dysfunction in Human lens epithelial cells	0.1, 1, and 10 μmol·L^−1^	Protocatechualdehyde alleviated Methylglyoxal (MGO)-induced mitochondrial dysfunction and apoptosis in human lens epithelial cells (SRA01/04 cells).	[151]
				Protocatechualdehyde was capable of inhibiting MGO-mediated advanced glycation end products (AGEs) formation and blocking receptor of AGEs expression in SRA01/04 cells.	

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
