# Peer review of "Biosynthesis, Chemistry, and Pharmacology of Polyphenols from Chinese Salvia Species: A Review"

_molecules, 2019, doi:10.3390/molecules24010155_

Round 1

Reviewer 1 Report

Correct title: Biosynthesis, Chemistry and Pharmacology of polyphenolsA from Chinese Salvia species: A review – delete A in polyphenolsA

Line 18: correct collation

Figure 1 correct Roamarinic acid

3.1. Phenolic acid should be written under one subtitle. It is not necessary to write so many subtitles. Authors should combine text and connect it.

Table 2: be careful with writing the name of compounds; names are not carefully split in another row

Line 164: correct [123]; where is reference 124

Table 2: make separations between activities; in this way it is not completely clear which activity is connected with ingredients, results and references

Author Response

Response to Reviewer 1 Comments

Dear Reviewer,

Thank you very much for your help with our paper (Manuscript ID: molecules-415061). We have revised our manuscript carefully based on each point raised in the review process. As suggested, the whole manuscript was improved. For your convenience, we have enclosed our revised manuscript where all changes remain marked in red.

The Responses to the reviewers' comments are as flowing:

Point 1: English language and style: English language and style are fine/minor spell check required.

Response 1: Thanks very much for reviewer comments, which are very helpful for us to improve the manuscript. As suggested, the whole manuscript has been re-proofread to ensure the best possible version. We hope the revised paper will be more clear and accurate on expressions.

Point 2: Correct title: Biosynthesis, Chemistry and Pharmacology of polyphenolsA from Chinese Salvia species: A review – delete A in polyphenolsA

Response 2: We are sorry for this mistake. We have deleted “A” in polyphenolsA. Please refer to page 1 line 3.

Point 3: Line 18: correct collation

Response 3: Thank you for your comments. We rechecked the word. We want to use the word to express the meaning of collection and arrangement. Please refer to page 1 line 18.

Point 4: Figure 1 correct Roamarinic acid

Response 4: We are grateful to the reviewers for pointing out our error. We have revised “Roamarinic acid” into “Rosmarinic acid”. Please refer to page 3 Figure 1.

Point 5: 3.1. Phenolic acid should be written under one subtitle. It is not necessary to write so many subtitles. Authors should combine text and connect it.

Response 5: Thank you for your comments. According to the suggestion, we have summarized the phenolic acids under one subtitle. Please refer to page 3-4 line 99-122.

Point 6: Table 2: be careful with writing the name of compounds; names are not carefully split in another row

Response 6: We are sorry for this mistake. We re-checked the format of the names involved in the table. Please refer to page 4-6 Table 1; page 9-11 Table 2; page 16-20 Table 3.

Point 7: Line 164: correct [123]; where is reference 124

Response 7:We are sorry for this mistake. We checked the order of references and marked them in the paper. Please refer to page 12 line 155.

Point 8: Table 2: make separations between activities; in this way it is not completely clear which activity is connected with ingredients, results and references

Response 8: Thank you for your comments. We make separations between activities. Please refer to page 16-20 Table 3.

We tried our best to improve the manuscript and made some changes in the manuscript. We appreciate for Editors and reviewers' warm work earnestly, and hope that the correction will meet with approval. Thank you very much for your comments and suggestions. Once again show my appreciation for your work.

Reviewer 2 Report

The manuscript submitted is a review dealing with biosynthesis, chemistry and pharmacological activities of different polyphenol derivatives present in Chinese Salvia species. Overall the article submitted seems interesting, on the other hand, several issues have to be addressed to improve the manuscript’s quality and its suitability for publication in high impact journal such as Molecules.

First of all, there is a large number of typos and minor writing inconsistencies which overall give the impression of sloppiness. For instance, the term ‘in vitro’ should be written in italic (line: 281, and in Tab 3).

Line 108: rosemary Latin name is Rosmarinus officinalis L. (also written in italic) not ‘Rosmarinus offinalis L.’.

Page 16: I think it should be Table 3, not Table 2, and please check tables and text for missing italic style for Salvia species (Tab. 3 description, lines 266, 268).

Also, Table 1: compounds #19, #22 ‘p’ as a para or ‘m’ as a meta also should be written in italic. Same with the cis prefix. Please check the rest of the text.

Table 1: what is the difference between compounds #37 (sagecoumarin [dimer]) and #58 (sagecoumarin [trimer])?

Table 2: compound #37 named quercetin: Quercetin doesn’t have glucose substituent (unless we're talking about its glycoside derivatives such as quercitrin or isoquercetin, but in that case compound #37 should not be named quercetin) nor any methoxy groups, only 4 hydroxyls. I would strongly recommend detailed checking of all structures descriptions in Table 2.

I would also consider renumbering compounds in Table 2. It’s getting messy in the text when there are two different compounds with the same number (for example compound #1 in Table 1 and compound #1 in Table 2 are two totally different structures) even though they belong to different classes of compounds.

In a conclusion, submitted article require minor revisions before its resubmission and eventual acceptance.

Author Response

Response to Reviewer 2 Comments

Dear Reviewer,

Thank you very much for your help with our paper (Manuscript ID: molecules-415061). We have revised our manuscript carefully based on each point raised in the review process. As suggested, the whole manuscript was improved. For your convenience, we have enclosed our revised manuscript where all changes remain marked in red.

The Responses to the reviewers' comments are as flowing:

Point 1: The manuscript submitted is a review dealing with biosynthesis, chemistry and pharmacological activities of different polyphenol derivatives present in Chinese Salvia species. Overall the article submitted seems interesting, on the other hand, several issues have to be addressed to improve the manuscript’s quality and its suitability for publication in high impact journal such as Molecules. First of all, there is a large number of typos and minor writing inconsistencies which overall give the impression of sloppiness. For instance, the term ‘in vitro’ should be written in italic (line: 281, and in Tab 3).

Response 1: We are grateful to the reviewers for pointing out our mistakes, which are very helpful for us to improve the manuscript. As suggested, the whole manuscript has been re-proofread to ensure the best possible version. We hope the revised paper will be more clear and accurate on expressions. According to the suggestion, we have italicized the whole text of the italic words (including the in vitro) in all italics. Please refer to page 2 line 50; page 3 line 104, 114; page 4-6 Table 1; page 8 line 133, 135; page 9-11 Table 2; page 12 line 162; page 14 line 258, 260; page 15 line 274; page 16-20 Table 3.

Point 2: Line 108: rosemary Latin name is Rosmarinus officinalis L. (also written in italic) not ‘Rosmarinus offinalis L.’.

Response 2: We are sorry for this mistake. We have modified “Rosmarinus officinalis L.” in italic form. Please refer to page 3 line 104.

Point 3: Page 16: I think it should be Table 3, not Table 2, and please check tables and text for missing italic style for Salvia species (Tab. 3 description, lines 266, 268).

Response 3: We are sorry for this mistake. We rechecked the tables and words modified them. Please refer to page 14 line 258, 260; page 16-20 Table 3.

Point 4: Also, Table 1: compounds #19, #22 ‘p’ as a para or ‘m’ as a meta also should be written in italic. Same with the cis prefix. Please check the rest of the text.

Response 4: We are grateful to the reviewers for pointing out our error. We have italicized the whole text of the italic words in all italics. Please refer to page 4-6 Table 1.

Point 5: Table 1: what is the difference between compounds #37 (sagecoumarin [dimer]) and #58 (sagecoumarin [trimer])?

Response 5: We are sorry for this mistake.  The “sagecoumarin” is caffeic acid trimer. Please refer to page 5 Table 1.

Point 6: Table 2: compound #37 named quercetin: Quercetin doesn’t have glucose substituent (unless we're talking about its glycoside derivatives such as quercitrin or isoquercetin, but in that case compound #37 should not be named quercetin) nor any methoxy groups, only 4 hydroxyls. I would strongly recommend detailed checking of all structures descriptions in Table 2.

Response 6: We are sorry for this mistake. We re-checked the names involved in the table. Please refer to page 9-11 Table 2.

Point 7: I would also consider renumbering compounds in Table 2. It’s getting messy in the text when there are two different compounds with the same number (for example compound #1 in Table 1 and compound #1 in Table 2 are two totally different structures) even though they belong to different classes of compounds. In a conclusion, submitted article require minor revisions before its resubmission and eventual acceptance.

Response 7: Thanks very much for the comments. We have renumbered the compounds in table 2. Please refer to page 9-11 Table 2.

We tried our best to improve the manuscript and made some changes in the manuscript. We appreciate for Editors and reviewers' warm work earnestly, and hope that the correction will meet with approval. Thank you very much for your comments and suggestions. Once again show my appreciation for your work.